# Emotional Experience and Feelings during First COVID-19 Outbreak Perceived by Physical Therapists: A Qualitative Study in Madrid, Spain

**DOI:** 10.3390/ijerph18010127

**Published:** 2020-12-27

**Authors:** Domingo Palacios-Ceña, César Fernández-de-las-Peñas, Lidiane L. Florencio, Ana I. de-la-Llave-Rincón, María Palacios-Ceña

**Affiliations:** 1Department of Physical Therapy, Occupational Therapy, Rehabilitation and Physical Medicine, Universidad Rey Juan Carlos, 28001 Madrid, Spain; domingo.palacios@urjc.es (D.P.-C.); cesar.fernandez@urjc.es (C.F.-d.-l.-P.); anaisabel.delallave@urjc.es (A.I.d.-l.-L.-R.); maria.palacios@urjc.es (M.P.-C.); 2Research Group of Humanities and Qualitative Research in Health Science of Universidad Rey Juan Carlos (Hum&QRinHS), 28001 Madrid, Spain; 3Research Group of Manual Therapy, Dry Needling and Therapeutic Exercise of Universidad Rey Juan Carlos (GITM-URJC), 28001 Madrid, Spain

**Keywords:** COVID-19, physical therapy, emotions, feelings, outbreak, qualitative research

## Abstract

Knowing the experiences and feelings of health professionals who have actively worked on the first-line during the first COVID-19 outbreak could help for identifying specific professional duties focused on health assistance objectives. No qualitative study has been published describing the emotion/feelings of physical therapists during the first COVID-19 outbreak. This study describes and explores the emotional experiences and feelings of thirty physical therapists working at the first-line at eleven public health hospitals in Madrid (Spain) during the first COVID-19 outbreak (March–May 2020). A qualitative exploratory study was conducted based on an interpretive framework. Participants were recruited by purposeful sampling and snow-ball techniques between May and June 2020. In-depth interviews and researchers’ field notes were used to collect the data. An inductive thematic analysis was conducted to identify significant emerging themes from verbatim transcription. After identifying 2135 codes and 9 categories, three themes emerged to describe their emotional experiences and feelings. First, “Critical events”, with negative and positive critical events. Second theme, “Emotional Roller Coaster”, with emotions, feelings, and coping strategies. Third theme: “Last words: Conclusions of the COVID-19 outbreak experience”, with the meaning of the COVID-19 outbreak from a personal and professional perspective. Comprehensive support for all first-line healthcare professionals is needed.

## 1. Introduction

The world is facing an unprecedented health crisis due to the severe acute respiratory syndrome coronavirus 2 (SARS-CoV-2), known as COVID-19, a new ribonucleic acid coronavirus identified in Wuhan, China in December 2019 [1].

A rapid spread of the virus around the world led the World Health Organization (WHO) to define COVID-19 as a global pandemic on March 11, 2020 [2]. Since then, COVID-19 has spread rapidly across the globe in an extremely short space of time and with catastrophic and epic proportions to the society.

Spain is one of the most affected European countries showing one of the highest confirmed COVID-19 cases (data November 25th 1,600,000 cases) [3]. The COVID-19 pandemic is representing one of the most significant challenges to worldwide healthcare systems. Due to the sudden spread of COVID-19 virus, healthcare professionals have experienced a huge, quick, and deep change of their professional work for battle against the epidemic and to care for individuals infected with COVID-19. This situation has led to a potential psychological and emotional impact in healthcare professionals. Several studies have documented the presence of mood disorders, e.g., anxiety or depression, in healthcare professionals during the first COVID-19 outbreak [4,5,6]. In Spain, the number of healthcare professionals infected is one of the highest worldwide (95,000 infected approximately) [7]. Luceño-Moreno et al. found that 56.5% of healthcare professionals working in Madrid (Spain) showed symptoms of post-traumatic stress disorder, 58.5% anxiety, 46% depression, and 41.1% feel emotionally drained during the first COVID-19 outbreak [8].

The relevance of healthcare professionals in the COVID-19 outbreak has been clearly demonstrated; hence, awareness of what these professionals had experienced during the first acute COVID-19 outbreak could be crucial for identifying situations where changes against future waves of the COVID-19 virus could be improved. Previous studies have mainly focused on the presence of mood disorders, e.g., anxiety or depression, by using specific questionnaires; however, few studies have investigated the emotions or feelings of these healthcare professionals [9]. Identifying the predominant emotions and feelings of healthcare professionals working in the front-line during the first COVID-19 outbreak, those events and situations provoking and promoting these emotions or feelings, and the potential copying strategies used for affronting these emotions or feelings, can help for implementation of preventive programs in healthcare professionals at a higher risk [10].

In such a scenario, qualitative research including personal narratives provides a personal picture of COVID-19 experience that may be highly meaningful to health care administrators. In fact, qualitative research describing the experience of those healthcare professionals involved in the first COVID-19 outbreak is increasing. Liu et al. explored the experience of nine nurses and four physicians during the first COVID-19 outbreak and observed the predominance of responsibility and resilience emotions when working on COVID-19 wards [11]. Sun et al. described the experience of twenty nurses and reported an evolution from negative to positive emotions during the first outbreak [12]. Similarly, Galehdar et al. described that nurses attending COVID-19 patients experienced a variety of psychological distress during the first outbreak [13]. Most qualitative studies have been conducted in a specific healthcare population, nursing. Ardebili et al. included nurses but also other health professionals, e.g., pharmacists, laboratory personnel, and radiology technicians [14].

It is important to note that all healthcare professionals, e.g., emergency clinicians, surgeons, internists, nurses, and physical therapists, have been actively involved working in the front-line during the first COVID-19 outbreak. Knowing the experiences of different healthcare professionals who actively worked against COVID-19 pandemic at the first outbreak is needed since all types of professionals have simultaneously fought against the virus; however, they have experienced different duties and their emotional experiences and feelings can differ from each other [15]. The authors have been unable to find previous qualitative studies describing the emotions/feelings experienced by physical therapists. Therefore, the purpose of this qualitative study was to explore the emotional experience and feelings of physical therapists working as the first-line in public health hospitals in Madrid (Spain) during the first COVID-19 outbreak.

## 2. Methods

### 2.1. Study Design

A qualitative exploratory study was conducted based on an interpretive framework [16]. This study was conducted according to Consolidated Criteria for Reporting Qualitative Research (COREQ) [17] and the Standards for Reporting Qualitative Research (SRQR) [18].

### 2.2. Ethics

The study was approved by the Local Ethical Committee of Universidad Rey Juan Carlos (URJC 1905202011920). All participants provided oral informed consent prior to their inclusion.

### 2.3. Research Team

Prior to the study, the researchers’ positioning was established via two briefing sessions addressing the theoretical framework for this qualitative study, their beliefs and their motivation for the research [19]. Five researchers (three women) participated in this study, including one anthropological nurse (DPC) and four physical therapists (CFdlP, LLF, MPC, and ALLR). All researchers have experience in research in health sciences and none were involved in related-clinical activity associated with the participants.

### 2.4. Participants, Context, and Sampling Strategies

Thirty physical therapists, aged from 18 to 65 years old, working in rehabilitation/physical therapy services at public health care hospitals in Madrid (Spain) during the first COVID-19 outbreak were recruited between 25 May and 30 June, 2020. The city of Madrid (population 6.6 million) has become one of Europe’s areas most affected by COVID-19 (350,000 COVID-19 cases on November 2020).

Participants were recruited through purposeful and snowball sampling techniques [20]. Twelve physical therapists working at 3 public health hospitals of the south-west area of Madrid providing direct care for patients with COVID-19 during the first outbreak were initially eligible. The remaining 18 physical therapists were recruited via snowball sampling from other eight public hospitals located at different areas of Madrid (north-west, north-east, south-east, and city center). Both sampling techniques help to expand the sample size and to improve the diversity of the participants. In qualitative research, the sample size is commonly determined by the data saturation; however, in the current study we used the proposal made by Turner-Bowker et al. who reported that 99.3% of concepts, themes, and contents emerged with around 25 interviews [21]. Turner-Bowker et al. proposal usually tends to increase the sample size when compared to data saturation [21].

### 2.5. Data Collection

Semi-structured, in-depth interviews including open questions were used to obtain information regarding specific issues of interest [16,20]. After collecting professional and personal data from each subject, the following broad opening question was conducted: “Please, can you share with me your personal emotional experience during the COVID-19 outbreak?” Open-ended follow-up questions were also used to obtain the detailed descriptions, some examples included: 1, “Do you consider that COVID-19 outbreak has changed your life?”; 2, “How do you feel when you were working with COVID-19 patients?”; 3, “How do you cope with this emotional situation in your work and life?”; 4, “What were your thoughts and feelings about the COVID-19 outbreak?”; 5, Are you able to describe your feelings during the COVID-19 outbreak?; 6, Are you able to describe your emotional experience of the COVID-19 outbreak with just one sentence? Additionally, “Please tell me more about that”, was also used during all the interviews (if needed) to enhance the depth of the discussion of a specific topic.

Due to the lockdown situation for flattening the COVID-19 curve established by the Spanish Government on 14th March 2020 [22], interviews were conducted at a private video chat room using the videoconference platform Zoom (www.zoom.us, San Diego, CA, USA) [23].

Each participant received a private/personalized email with an invitation to the interview (Table 1 shows the specific procedure followed for the interviews using the zoom platform).

With participant oral permission, all interviews were audio- and video-recorded to get access to non-verbal cues such as eye contact, facial expressions, or body motions, unique data resources for qualitative studies. Video-recording permitted us to collect as much non-verbal information as possible, which could enrich the descriptions of health care providers’ experiences. All interviews were transcribed verbatim, recording a total of 1434 min of interviews overall (average of 47.8 ± 12.8 min each interview).

Additionally, field notes during the semi-structured interviews were also collected by the researchers since field notes provide a rich source of information as participants describe their personal experiences, their behaviors during data collection, and enable to note their reflections concerning methodological aspects of the data collection [19].

Confidentiality was assured by consecutively numbering each interview and removing identifying information from the transcripts. All audio recordings and transcripts were saved on a password-protected computer with restricted access only by the researchers and deleted one month after the analysis.

### 2.6. Data Analysis

The full literal transcription of each interview, the researchers’ field notes and their descriptions were all collated to perform a qualitative analysis [21,24]. A thematic, inductive analysis was performed [24,25]. The thematic analysis consisted of identifying the most descriptive content to obtain codes, and subsequently reduce and identify the most code groups (categories). Accordingly, groups of meaningful codes were formed (i.e., similar points or content that allowed the emergence of the topics that described the participants’ experience) [24,25]. This thematic analysis process was separately done upon the semi-structured interviews. Subsequently, joint meetings were held to combine the results of the analysis, to represent the participants’ experience [24,25].

In cases of potential discrepancies, theme identification was based on establishing a consensus between the research team members. No qualitative software was used to analyze the data.

### 2.7. Rigor

The techniques performed and application procedures used to control trustworthiness are described in Table 2 [26].

## 3. Results

Thirty physical therapists (63% women, mean age 41 ± 6 years, mean 20 ± 7 years of clinical experience) working in 11 public hospitals of Madrid were enrolled. Twenty-eight (93%) worked as clinicians, whereas two (7%) had management positions at the rehabilitation/physical therapy services. All participants have actively worked with COVID-19 patients in the hospitals in which they were employed during the first COVID-19 outbreak from mid-March to the end of May 2020.

The data analysis revealed 2135 codes and 9 categories. In addition, three main themes were identified: (a) critical events, with two categories; (b) emotional roller coaster, with two categories; and (c) last words: conclusion of the COVID-19 outbreak experience, with five categories (Table 3).

Below, we included some of the patients’ narratives taken directly from the interviews in relation to the three emerging themes and subthemes.

### 3.1. Theme 1. Critical Events

During the first COVID-19 outbreak, the rehabilitation services were closed. Physical therapists had to work within pharmacy services (making hydroalcoholic gel, transporting medication to chronic patients to their home), on intensive care units—ICU (helping in mobilization and pronation), in the warehouse, in the preventive medicine service (helping to reduce contagion), and in the sterilization department (making and delivering protective personal equipment—PPE). This situation exposed professionals to different work environments during the COVID-19 outbreak.

Physical therapists narrated how they lived through critical events during the outbreak. These critical events were situations that, due to their frequency, their harshness, and/or drama, impacted all the participants. Critical events crumbled and sank some of the participants, while others encouraged and strengthened them.

#### 3.1.1. Negative Critical Events

Participants identified the following negative critical events as their experience during the COVID-19 outbreak: 1, the death of patients while helping other patients; 2, all members of a family entering the hospital and only one survive; 3, the family took a loved one to the emergency room and minutes later they were informed of his/her death; 4, patients who died during the visit of their relatives; 5, patients who died alone; 6, families who could not accompany their loved ones and were notified by telephone of their death; 7, professionals without PPE who continued to work in the first week of the outbreak; 8, healthcare personnel maintained ventilatory support for patients for hours (without respiratory equipment) until they were exhausted; 9, participants experiencing contagion, hospitalization, and death of comrades; 10, there were not enough respirators for ICU patients; 11, some patients treated by participants who died progressively; 12, the fragility and dependency exhibited by COVID-19 survivors (several could not even move their hands); 13, when you finish prone position of a patient for waiting for her death; 14, the incredulous eyes of isolated patients; 15, the death of young patients without a risking profile; and, 16, the loneliness of isolated patients. Here, we reflect some literal narratives from participants:
“.. the worst was not the fear of contagion, but the daily contact with the death.”(p3)
“For me it was a conflictive experience, you tried to be a professional and continue treating patients. But you knew that the next day they could have died.”(p12)
“The experience of the COVID-19 was brutal, without material, without space, people in the corridors, one person died, healthcare professional removed her/him and two minutes after another patient occupied the same bed.”(p16)
“The situation with a huge impact to me was is the human suffering. The numbers have not reflected the truth.”(p17)
“You faced the death each day, your patients were continually dying, you worked with the dead around you. You go to another room, and, more dead, patients lying covered with a sheet waiting to be taken to the mortuary.”(p23)
“When I entered to the library of the hospital, my breath stopped. The library was used to sedate patients who were going to die. I saw screens covering people who had died and were waiting to be picked up… I will never forget that picture.”(p27)
“I was really impressed when they said me that you had to “release the prone position” of a patient to release.” “This meant that the patient was going to die, and they released him from the prone position, placing him on his back, more comfortable, to die.”(p17)
“I will not forget the eyes of the patients. A look of disbelief, of not believing what is happening. The eyes of people being on the verge of death, and perceiving fear in their eyes, without being able to communicate because they cannot speak.”(p21)

#### 3.1.2. Positive Critical Events

On the other hand, participants also identified the following positive critical events: 1, mutual support and reinforcement among healthcare professionals; 2, developing of a huge capacity for sacrifice and strength of healthcare professionals to work hours and hours; 3, human quality of healthcare professional to treat patients; 4, the continuous monitoring of COVID-19 patients, who were alone for weeks/months; 5, the dedication and effort of all physical therapy service to adapt for working in unknown duties into the hospital; 6, the corporative union of physical therapists working all together; 7, the commitment of healthcare professionals with the patients and with their hospitals; 8, the hospital’s ability to mobilize all its resources in less than 3 days and prepare the hospital for the acute COVID-19 outbreak; 9, the capacity of hospitals and health system to be able to receive as much as patients as possible; 10, the ability to open fast new ICUs and hospitalization wards; and, 10, to be proud of being a healthcare professional and having been able to help during the pandemic. Again, some literal narratives from participants were:
“For me it was incredible how everyone offered and helped. You felt that you were not alone.”(p24)
“Working in the hospital was working on the front-line of the war, you always felt overwhelmed, you go to work with fear, you cry and cry, but whatever happened, you were in the front-line against the virus.”(p5)
“The group of physical therapists, we worked together, in total synergy. We supported each other and tried to stay together against the adversity. The COVID-19 outbreak destroyed everything, your hope, your spirit.”(p14)
“It was incredible, the touch, the respect with which everyone tried to treat patients. They were extremely fragile; we didn’t want to harm them in any way.”(p5)
“Many times, at the end, you went to see or visit a patient. Typically, COVID-19 patients were alone. When I could, I tried to prolong my treatment to spend more time with them.”(p22)
“None hospital was ready for this. But if I’m honest, they adapted extremely fast, and, in two days, the entire hospital was transformed to deal with the COVID-19. I still don’t believe it.”(p19)
“I don’t know what other countries have done, but I have the feeling that here, no one was left behind. All patients were treated and received at the hospital. For me, it was an honor to be part of this.”(p16)

Some participants narrated how several events remained in their minds as static images and photographs, rather than words. They did not recall a particular sequence in their memory. They have scenes that reflected a moment on his memory. For instance:
“Memories are like frames, images, not a sequence like a DVD movie, only images of what happened.”(p3)

### 3.2. Theme 2. Emotional Roller Coaster

This theme describes the emotions and feelings of participants during the COVID-19 outbreak, the situations that could provoke the emotions and the coping strategies that they used. The emotions/feelings were highly heterogeneous and variable, therefore, the majority of participants described it as “a roller coaster” or “a storm” of emotions and feelings.

#### 3.2.1. Emotions and Feelings

The most narrated and shared emotion experienced by all participants was fear. The moments in which they were afraid included: 1, fear of being infected by the virus; 2, fear of developing the disease; 3, fear of being admitted into the hospital; 4, fear of being admitted to the ICU; 5, fear of needing a respirator; 6, fear that, if they need, there were not enough respirators; 7, fear of spreading the disease to their colleagues; 8, fear of spreading the virus to home; 9, fear of making mistakes when putting on/taking off the PPE; 10, fear of working in first-line against the virus (i.e., emergency, ICU); 11, fear of going to work at the hospital; 12, fear of that the material disappear; 13, fear of leaving their children orphans because they died due to COVID-19; and, 14, fear of that the PPE would be of poor quality and they could be infected.

Participants clearly reported how the fear of being admitted to a hospitalization was different from the fear of being admitted to the ICU because being admitted to the ICU means a worse prognosis, more severe disease, and uncertainty of having resources such as a respirator.

“It was different. Entering an ICU was very scary, especially when you know that respirators may not be available for you.”(p20)

Some participants describe fear with capital letters, not a specific emotion, the fear does not finish after work, this emotion remained and accompanied the professional to their home, always.

“You take fear home with you, even shopping, you have to force yourself.” (p8). “I do not remember having a particular moment that was mostly scared and marked me. The fear started with the first day of outbreak, it was like a constant horror movie. I’m still inside, I’m still inside, I’m still in “pandemic mode.” It is having constant fear.”(p29)

Participants described how fear did not paralyze them, but it was always there, like a signal. “Fear could not block you, you kept active working, but it was a reminder, a warning that you always had to take into consideration.” (p4).

Additionally, other emotions and feelings included anger, sadness, stress, frustration, helplessness, indignation, anguish, anxiety, sadness, illusion, rage, uncertainty, pride, disbelief, panic, worry, loneliness, rage, grief, hope, joy, or gratitude. Some participants described experiencing a “roller coaster of emotions”, an “emotional chaos”, a “storm of emotions”, or with “emotions on the skin”.

“You felt a lot of nerves and fear always. Other times you didn’t know if you were sick, it was just annoyances, or you had COVID -19. Others had moments of loneliness and suffering. A mess.”(p14)

“… It was living in a storm of feelings… positive and negative feelings, of all colors.”(p20)

This emotional roller coaster exhausted the majority of participants. “Everything was very intense. We lived everything very extremely. I think it was justified what we felt. Around us, there was a lot of death and personal darkness. I ended up exhausted, I suffered a lot.” (p14).

#### 3.2.2. Coping Strategies

Participants reported various strategies for coping with their emotions and feelings. They did not confirm whether these strategies followed a professional recommendation. All participants agreed that at the beginning of first COVID-19 outbreak, there was no professional psychological help for healthcare workers. Only one participant confirmed that she received professional psychological therapy.

Some participants described how they did not share their feelings enough. They believed that in the future many healthcare professionals would experience psychological problems, particularly those who had worked on the front-line against the virus. Many participants agreed that all health professionals have been emotionally touched.

“I prefer not to talk to anyone; people would not understand. You must have lived it. I don’t want to worry or scare my family.”(p27)

Most participants narrated how their way of dealing with their emotions/feelings was to rely on all their healthcare colleagues.

“When I was home in isolation, I received messages from all over the world. I felt very supported.”(p15)

Participants reported how they used work as a tool to manage their emotions of fear or anger. They did not stop working, they kept doing their job the best they could, and they did not want to think of anything else.

“The first thing he did when he arrived to work was cry, he finished and was ready to work, another day.”(p20)

For other participants, it helped not to think of patients as real people, with a life and a family.

“You don’t think they are real and live people, with family, children, and a life. Otherwise, you could not continue working. Note that in a 7-h work schedule, up to 7 patients could die.. I suffered too much.”(p17)

In addition, some participants reported how they became insensitive to the pain of other people in several situations. It was something they could not control, it just happened.

“One day the father of a colleague was dying, and they called her to say him goodbye. At that moment I didn’t feel anything, I didn’t realize what I was experiencing. I realized after two days.. when I realized it, I burst into tears.”(p24)

Other participants faced the experience by focusing on what they have been able to contribute during the outbreak. They did not look back and only looked forward, projecting into the future what their lives will be like after the pandemic. The pandemic has been a stage in their life, which should not affect the rest of their life.

“I want to focus on the journey I have made. I am not going to focus on specific things, I am not going to look back, I am only going to have a prospective vision.”(p3)

“I am not going to let the COVID-19 condition my whole life. The COVID-19 outbreak has been a moment in my life, nothing more. I have a lot of things to live.”(p23)

In those participants with fear of working in the front-line, the fear diminished as more experience and security were gained with the virus. Those afraid of being infected, took an extreme hygiene routine, took their temperature continuously, or stayed away from people and family.

“My head went off, I got a psychosis from washing everything, disinfecting everything, I didn’t touch anything but I had gloves.”(p15)

The majority of participants confirmed that it was a moment of self-assessment and self-knowledge. In fact, humor sometimes helped them to fight against the critical events and overcome adversity.

“I know it was wrong to joke right now. But sometimes it was a relief, an evasion of all the human drama that was around the outbreak.”(p4)

### 3.3. Theme 3. Last Words: Conclusions of the Outbreak Experience

This theme described the participants’ responses to the question: “Are you able to describe your experience of the COVID-19 outbreak with just one sentence?”. Not all participants were able to describe the situation.

#### 3.3.1. A Contradictory Experience

Six participants described that the experience as contradictory because, despite human drama lived during the outbreak, they have had a positive or constructive feeling of that experience. These participants reported how they have had a professional and personal growth during the first COVID-19 outbreak.

“I have conflicting feelings because people were suffering, but it was also a potential opportunity for physical therapy, as a science, to be able to advance and be able to help more people in the future.”(p7)

“The summary of the experience; enriching. On one hand it was a real tragedy, but on the other, it was enriching, because I was immersed in an interdisciplinary team with other professionals, working all together.”(p12)

“Everything was horrible, but at the same time I felt better than ever in my life, in my work. I feel 200%.”(p15)

The majority of participants described a deep personal change, their lives, their values, and their priorities have changed, something has awakened inside them, their “world has changed” and it will never be the same again.

“In conclusion, I have changed as a person. I’m not the same. Everything has changed. But not everything was bad. I have learned a lot as a professional and as a person.”(p9)

The expressions that participants used to describe their experience during the COVID-19 outbreak summarizing what they lived have different meanings and can be divided into two main blocks. Fourteen participants used “affection and recognition”, “learning”, “overcoming”, “personal and professional growth”, “enriching”, “hope”, “strength”, “learn to prioritize and stop”, “metamorphosis”, “encourage”, “respect and transcendence”, “rewarding”, “pride”, “opportunity”, “extremely positive experience”, and “resistance”. Six participants used: “wild”, “loneliness”, “fragility”, “suffering”, and “intense and exhausting” experience.

#### 3.3.2. The Meaning of the Relationship with Patients during the Outbreak

“Respect” meant acceptance of the decisions and needs of COVID-19 patients. “Transcendence” meant helping and accompanying the person throughout their process. “The affection” of the patients and their families towards healthcare professionals was the way of recognizing “the human quality” of their work.

On the other hand, “loneliness” meant the isolation of the patient, perceiving his suffering from the other side of the window, perceiving the patient’s loneliness when they were ill and dead. It also meant the loneliness of the professional during the human drama of the pandemic. “Frailty” meant the vulnerability of people to an unpredictable, an invisible, and very small enemy.

“The most positive and precious thing during the outbreak were the video calls done with families. Everyone connected, you saw the relatives and the patient crying and laughing.. people felt grateful and they shared this emotion with you. They continually recognized your work.”(p28)

“The most brutal thing was loneliness… The loneliness of the patient, the loneliness of the healthcare professional looking impotent because we couldn’t save everyone … The fragility of people. Because of a bug that you cannot see, which is insignificant.”(p14)

#### 3.3.3. The Meaning of the Relationship between Physical Therapists and the Remaining Professionals during the Outbreak

The relationship between participants and other healthcare professionals was experienced as “enriching and positive”, because they shared knowledge and techniques, and it allowed physical therapists to get out of their comfort zone (rehabilitation service). “Fellowship” means to face adversity together, to work as a real team, and to support one another. “The resistance, courage and strength” meant facing the death of COVID-19 patients, the lack of resources and PPE, and continuing to work, day after day. Most participants perceived “a metamorphosis” at their professional level. The pandemic has tested their capabilities and they grew as professionals. On the other hand, “intense” meant living with everything being very extreme and exhausting. The COVID-19 outbreak was a situation of life or death, and it was highly demanding. This caused physical, emotional, and spiritual “exhaustion”, an “emptying” of energy, with a potential losing of purpose of your contribution as a physical therapist.

“There has been a change in all professionals, a great camaraderie, we all have the feeling of having won something together.”(p12)

“There were days when you couldn’t do anything more, it wasn’t just physical, it was more internal, like a loss of sense of everything.”(p23)

#### 3.3.4. The Meaning of the Outbreak at a Professional Level

At a professional level, a “rewarding” experience meant being able to participate and help people. Professional “pride and growth” have been perceived as an opportunity to demonstrate how physical therapy profession can help and contribute to the hospital. “Opportunity and hope” meant being able to start things from a beginning, change what does not work, implement new proposals, and participate as a healthcare professional in the pandemic.

“This has been an opportunity to start from the beginning, doing things as well as you could. It’s a shame that a pandemic had to happen to do thing well.”(p17)

#### 3.3.5. The Meaning of the Outbreak on a Personal Level

“Personal growth” meant to be able to overcome personal fears, to perceive the vulnerability of others, to accompany patients in extreme situations, and to recover their vocation as healthcare professional. “Learning to prioritize and stop” meant to give value to small things and people around them and to be able to reflect on their relationships with other people and their environment. On the other hand, “savagery” meant living dramatic moments and critical events every day. The magnitude of death and suffering was not acceptable at all. Everything was disproportionate, uncontrolled, and extreme.

“In summary, 15% bad and 85% good, with a lot of personal and professional growth. I’m going to fight to do a good job, not a substitute .. I’m a better person than before.”(p10)

## 4. Discussion

Our results showed how the presence of critical events marked the experience of physical therapists during the first COVID-19 outbreak. In addition, a great variety of emotions and feelings emerged like a storm/roller coaster, which was difficult to manage and cope with. Physical therapists narrated how they lived contradictory experiences, since they were able to perceive positive aspects within the drama of the pandemic, how they gave a positive meaning to the relationships with patients and other professionals, and how the COVID-19 outbreak impacted them at the professional and personal levels (Figure 1).

Health professionals had experienced, in general, a wide range of emotions during the unfolding of the COVID-19 pandemic [14]. The COVID-19 pandemic has had a massive and international impact on health care systems worldwide, and has also increased the risk of psychological distress in healthcare professionals [27]. Previous studies have shown that frontline healthcare providers treating COVID-19 patients experience higher risk of anxiety, depression, and insomnia [28,29]. Physical therapists, as healthcare professionals who have worked at the front-line of the COVID-19 outbreak, may also exhibit psychological distress [9,30]. No qualitative study has previously investigated this topic.

This is highly important since high levels of stress, anxiety or depressive symptoms could have long-term psychological implications in health professionals [28]. Previous studies reported that the emotional demands from healthcare professionals during the COVID-19 outbreak were positively associated with fatigue, depression, anxiety, and distress [31]. Zhang et al. showed that Chinese healthcare professionals experienced negative emotions including tense, scared, angry, sad, afraid, impressed, or confident during the COVID-19 outbreak [32]. Giusti et al. [27] showed that Italian health professionals different than physical therapists showed a high level of burnout, psychological symptoms, and emotional exhaustion during COVID-19 pandemic. It seems that prolonged exposure to an acute virus infection and intense work are detrimental factors for the mental health of healthcare professionals.

Previous studies had also shown how healthcare professionals during the COVID-19 outbreak exhibit alternance between negative and positive emotions [14,33]. On the one hand, overwhelming, ambiguity feelings, feeling helpless, feelings of pointlessness, guilt, and remorse, losing control, and a sense of providing futile care were identified when health professionals fought against events such as unprecedent heavy workload, depletion of existing human resources, shortage in protective devices, or patients die despite hard work. Other feelings also included anxiety against the death, anxiety due to the nature of an unknown virus, anxiety caused by corpse burial, fear of infecting the family, distress about time wasting, distress of always delivering bad news, fear of being contaminated, the emergence of obsessive thoughts, and the bad feeling of wearing PPE [13]. On the other hand, positive feelings including conscientiousness and self-sacrifice for patients were also identified when healthcare professionals risking their health and lives for their patients and being loyal to the medical oath. The current study found that physical therapists also described their emotions experienced as a roller coaster during the first COVID-19 outbreak.

In Spain, risk factors associated to psychological distress and the presence of intensive and severe negative emotions in healthcare professionals included the lack of enough PPE under working exposure, non-existent or contradictory work protocols, working 12–24 h, or being worried that a family member could be infected by the virus [5,8,33]. Moreover, social isolation, uncertainty, reluctance to work, or considering absenteeism were also frequently reported by Spanish healthcare professionals [28]. Obviously, a global lockdown situation that healthcare professionals also lived in parallel at home could also have contributed to these emotions or feelings. In fact, anger, fear, sadness, disgust, and uncertainty were emotions reported by Spanish citizens during their lockdown at the first COVID-19 outbreak [34]. Zhang et al. [35] also reported that health professionals (nurses) had several negative emotional reactions in the early stages of home isolation. Therefore, it seems that healthcare professionals, independent of the profession, experienced a double stress situation, professional overload and personal/familiar lockdown, during the first COVID-19 outbreak.

Ardebili et al. [14] also showed how emotions and feelings were highly conditioned by the time of evolution of the pandemic. These authors developed a three-level model: early exposure, with high levels of anxiety and fear, powerlessness, and losing control; crisis peak, featured by feelings of futility, helplessness, hopelessness, frustration, emotionally exhaustion, and high feeling of losing control; and, long-term effects, with persistent fear of potentially getting infected, symptoms of post-traumatic stress disorder, feeling sad, and reactivation of earlier fears as patients number resurge. It is important to consider that the current qualitative study covered the early and crisis peak moments of the first COVID-19 outbreak. It would be interesting to describe emotional and feeling experience of physical therapists, and other healthcare professional, at the long-term moment.

The results of the current study had consistently shown that physical therapists also experienced similar emotions and feelings as other healthcare professionals, since fear was the most intense and long-lasting emotion in these professionals [14,28,36]. During the first COVID-19 outbreak, various types of fear have been identified [37]; (a) the fear of attendance at work (doctors and nurses avoid accepting patients in the hospital and staff avoid being back at the hospital), (b) fear of infection (concern about family and the virus), (c) fear of getting sick (themselves or their family), (d) fear of more COVID-19 waves, (e) fear of an increase in mortality (death of patients and colleagues, particularly medical colleagues’ death had a huge negative effect on health professionals). Ardebili et al. [14] and Barello et al. [28] showed that healthcare professionals who worked during the pandemic reported concerns regarding their fear of dying alone and being separated from their loved ones, fear of infecting their families, friends, or colleagues, and fear of stigmatization. Again, the results of this study including physical therapists revealed similar experiences to other healthcare professionals such as nurses or medical doctors.

Previous studies have also shown how healthcare professionals felt stigma, rejection, and even loneliness from their neighbors and social environment because of working in a hospital [28,36,38]. Our participants have not reported the presence of stigma in their family and social environments. It is probably that this emotion would be more present in other healthcare professions, e.g., nurses. This discrepancy may be related to the fact that the physical therapist is a healthcare professional not usually involved in daily functions of saving lives or in daily contact with potential infectious patients at their regular hospital work. This is mostly associated with nurses or medical doctors, since these healthcare professionals have completely different roles in hospital daily work than physical therapists. In fact, the COVID-19 outbreak changed completely the function of physical therapists at the hospitals, since these professionals passed from their usual location at their rehabilitation services to several other places. As it was reflected in the theme 1 about critical events, rehabilitation services were closed and physical therapists were derived to completely different services at hospital, i.e., pharmacy services, sterilization department, or ICU.

Our results were also consistent with previous studies reporting that other medical professions exhibited changes in their daily life and relationships with loved ones during and after the COVID-19 outbreak [14]. With the passage of time, the pandemic situation had become the normal life for healthcare professionals since they have normalized and adapted to the virus. The meaning of this normalization consists of obtaining experience with virus treatment knowledge and skills, restoration of self-confidence in themselves, learning to live with the disease and using protective measures as a new way of life, and adopt new life styles. It seems that the COVID-19 virus has and will change our life independently of the healthcare profession.

Previous studies have also reported that healthcare professionals did not receive professional psychological support from their workplace centers during the first months of the acute COVID-19 outbreak [36,37,38]. In Spain, Martínez-López et al. reported that 90% of the healthcare professionals considered that psychological care should have been provided from their workplace centers [33]. Importantly, 43.3% of health professionals estimated that they would need psychological treatment in the near future. It would be expected that the psychological impact of COVID-19 will be long-lasting and the society worldwide will need to take the appropriate sources for this sequela in healthcare professionals [39]. It has been suggested to research preventive psychological support and to guarantee reasonable work conditions to protect healthcare professionals from these long-lasting psychological effects of the first COVID-19 outbreak [28].

Resilience is one of the strategies to manage healthcare professionals’ feelings and emotions and to cope with stress [40]. Resilience can be defined as the individual’s ability to deal with adversities as it is able to reduce the impact of post-traumatic stress, and manage emotions/feeling such as fear, sadness, frustration, irritability, guilty, and anger. Heath et al. described self-care, organizational justice, individual, and organizational strategies as the management approaches to increase resilience in healthcare workers during the COVID-19 pandemic [40]. Nevertheless, interventions should not be generic but should rather be tailored-made to an individual’s personal situation and characteristics [14].

Additionally, personal resources for reducing distress during the pandemic such as problem-oriented coping, favorite personal things, compassion satisfaction (empathy with the patient, joy of improving patients’ health, interest in people, and commitment to the patient), spirituality (strong faith, hope God’s grace and pray), personality traits (high self-confidence, challenging interest, realism, sense of humor, adaptability, hope, courage, strong thinking, adherence to ethics, relax, and flexibility), and social support are strategies used by healthcare professionals [38]. Finally, having the necessary physical sources for their protection and to update regular and accurate information is also essential for avoiding feelings of fear and uncertainty within the health professionals. This would improve the mental health of healthcare professionals [33].

The results of this qualitative study should be considered, attending to its potential strengths and limitations. Previous studies reported emotional experiences from nurses or other professionals such as prehospital emergency services, physicians, pharmacists, laboratory personnel, radiology technicians, hospital managers, and health managers in the ministry [9,13,14,27,37,41]. However, to the best of the author’s knowledge and based on a deep literature search, this is the first qualitative study describing the emotions and feelings of physical therapists who had worked at the front-line with COVID-19 during the first outbreak in public hospitals in a European Country. It is interesting to note that the emotional and feeling experiences reported by Spanish physical therapists share similar contents with other healthcare professionals from other countries. According to qualitative research international guidelines [21], the included sample size provides an in-depth insight (99% of the potential themes) of the potential participants’ experience. Another strength would be the inclusion of healthcare professionals from eleven different hospitals distributed throughout the Madrid region, one of the most affected areas during the first COVID-19 outbreak in Europe.

Among the limitations, due to the sanitary situation, interviews were conducted with an online digital platform, which may have reduced personal interaction between participants and researchers. However, online digital platforms permit visual feedback, which is not possible by using other telerehabilitation procedures such as a telephone [11]. In addition, our results cannot be extrapolated to all physical therapists or other healthcare professions who have worked during the first COVID-19 outbreak, due to the qualitative design. Nonetheless, these results may help to understand physical therapists’ experiences and their difficulties in managing and coping their feelings and emotions during the COVID-19 pandemic.

## 5. Conclusions

Our results provide insight on how emotions and feelings are experienced by physical therapists during the first COVID-19 outbreak, which may be helpful to organize and develop specific coping strategies for these particular healthcare professionals. In future studies, it would be necessary: (a) to describe the presence of emotions and feelings in other healthcare professionals who have not directly worked on the frontline of the COVID-19 outbreak; (b) to analyze the coping strategies used by different healthcare professionals to manage their emotions/feelings; and (c) to long-term monitor the long-term emotional health status of healthcare professionals who had worked during the first COVID-19 outbreak at the front-line.

## Figures and Tables

**Figure 1 ijerph-18-00127-f001:**
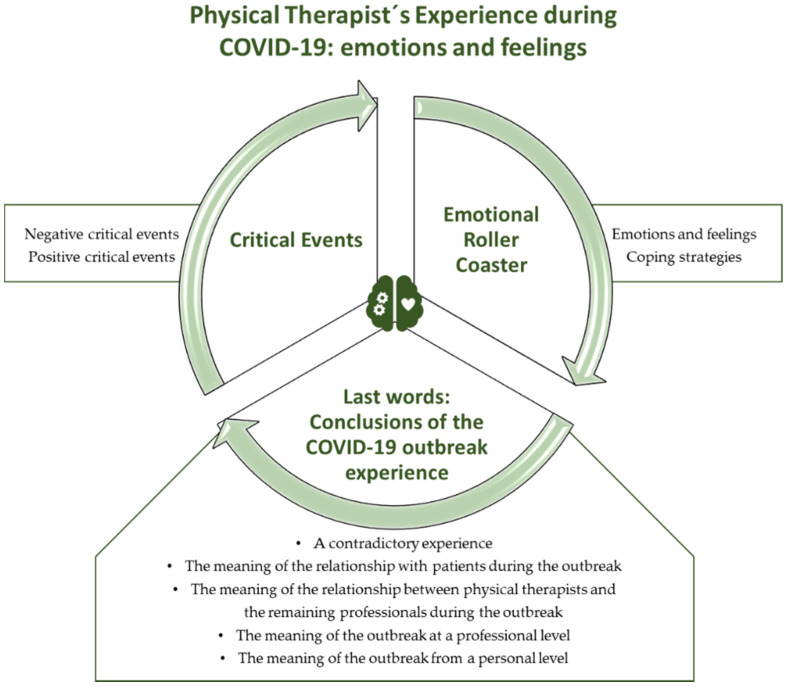
Emotional experience of physical therapists during the first COVID-19 outbreak.

**Table 1 ijerph-18-00127-t001:** Interview procedure by using Zoom Enterprise Video Communications (www.zoom.us, San Diego, CA, USA).

At the prearranged date and time, the participant and researcher both clicked on the zoom link and entered into the private video chat room. The interview involved the researcher first sharing the screen with the participant and reviewing an informed consent form together, reading and ensuring participant comprehension. After verbal consent to participate was provided, all participants were offered an email copy of the consent form. The researcher asked for participant permissions for recording the interview in both video and audio, and after confirming with the participant, researcher turned on the recording. If a participant declined to record a video, only audio was obtained The researcher asked the participant their personal and professional data. The researcher opened the semi-structured interviews guide (in the Microsoft Word document) on his computer and starting the interview with the opening question. The researcher asked participants to tell his/her about themselves, their emotions and feelings during the COVID-19 outbreak, to obtain a better understanding of how their unique situation may affect their comprehension or interpretation of its interview. During the interview, the researcher took notes on their responses. At the end of the interview, the audio/video was stopped after obtaining the participant consent.

**Table 2 ijerph-18-00127-t002:** Trustworthiness criteria.

Criteria	Techniques Performed and Application Procedures
Credibility	Investigator triangulation: each interview was analyzed by two researchers. Thereafter, team meetings were performed in which the analyses were compared and categories and themes were identified. Participant triangulation: the study included participants belonging to different hospitals. Thus, multiple perspectives were obtained with a common link (the emotional experiences and feeling of physical therapists during the first COVID-19 outbreak). Triangulation of methods of data collection: semistructured interviews were conducted and researcher field notes were kept. Participant validation: this consisted of asking the participants to confirm the data obtained at the stages of data collection. All participants were offered the opportunity to review the audio and/or video records to confirm their experience. None of the participants made additional comments.
Transferability	In-depth descriptions of the study performed, providing details of the characteristics of researchers, participants, contexts, sampling strategies, and the data collection and analysis procedures.
Dependability	Audit by an external researcher: an external researcher assessed the research protocol, focusing on aspects concerning the methods applied and study design. Additionally, an external researcher specifically checked the description of the coding tree, the major themes, participants’ quotations, quotations’ identification, and themes’ descriptions
Confirmability	Investigator triangulation, participant triangulation, and data collection triangulation. Researcher reflexivity was encouraged via the performance of reflexive reports and by describing the rationale behind the study.

**Table 3 ijerph-18-00127-t003:** Summary of the main themes and categories identified.

Themes	Categories
Critical events	Negative critical events Positive critical events
Emotional Roller Coaster	Emotions and feelings Coping strategies
Last words: Conclusions of the COVID-19 outbreak experience	A contradictory experience The meaning of the relationship with patients during the outbreak The meaning of the relationship between physical therapists and the remaining professionals during the outbreak The meaning of the outbreak at a professional level The meaning of the outbreak from a personal level

## Data Availability

The data presented in this study are available on request from the corresponding author. The data are not publicly available due to ethical and legal reasons to protect participant identity.

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
