# Peer review of "Emotional Experience and Feelings during First COVID-19 Outbreak Perceived by Physical Therapists: A Qualitative Study in Madrid, Spain"

_ijerph, 2020, doi:10.3390/ijerph18010127_

Round 1

Reviewer 1 Report

The authors research the experience and emotional response of physical therapists in the pandemic of COVID-19 with a sound Qualitative Study design. 

I appreciate most content of the study to help me to understand the first line experience in the COVID-19 pandemic. I had a critical command for the interpretation and discussion. 

As the study focuses on physical therapists, it is necessary to explore whether they had a characteristic experience or response to the COVID-19. Their specific role in the hospital system or their role in their original work may make them experience a different response in comparison to other staff, such as the first-line emergence department team. I would suggest the authors should discuss the effect of "physical therapists" role in their response to this crisis. 

Author Response

Response Letter manuscript IJERPH-1036271

Emotional Experience and Feelings during first COVID-19 Outbreak perceived by Physical Therapists: A Qualitative Study in Madrid, Spain

We thank the reviewers for their comments. We would like to make the Editor know that we addressed suggestions made by the reviewer one and answered those from reviewer two who it seems has not understood that this is qualitative, and not quantitate, study. We have included a new figure summarizing the themes found in the study which we believe will help reader. All changes are highlighted to facilitate the review process.

Reviewer 1

The authors research the experience and emotional response of physical therapists in the pandemic of COVID-19 with a sound Qualitative Study design. 

Response: We would like to thank for this positive feedback.

I appreciate most content of the study to help me to understand the first line experience in the COVID-19 pandemic. I had a critical command for the interpretation and discussion. 

Response: We would like to thank for this positive feedback.

As the study focuses on physical therapists, it is necessary to explore whether they had a characteristic experience or response to the COVID-19. Their specific role in the hospital system or their role in their original work may make them experience a different response in comparison to other staff, such as the first-line emergence department team. I would suggest the authors should discuss the effect of "physical therapists" role in their response to this crisis. 

Response: We have included the following comment on this topic on lines 533-545:

“Previous studies have also shown how healthcare professionals felt stigma, rejection and even loneliness from their neighbors and social environment because working in a hospital [29,39,40]. Our participants have not reported the presence of stigma in their family and social environments. It is probably that this emotion would be more present in other healthcare professions, e.g., nurses. This discrepancy maybe related to the fact that the physical therapist is healthcare professional not usually involved in daily functions of saving lives or in daily contact with potential infectious patients at their regular hospital work. This is mostly associated with nurses or medical doctors, since these healthcare professionals have completely different roles in hospital daily work than physical therapists. In fact, the COVID-19 outbreak changed completely the function of physical therapists at the hospitals, since these professionals passed from their usually location at their rehabilitation services to several other places. As it was reflected in the theme 1 about critical events, rehabilitation services were closed and physical therapists were derived to completely different services at hospital, i.e., pharmacy services, sterilization department, or ICU.”

We hope that the current version of the manuscript can get a positive review and can be accepted for publication in International Journal of Environmental Research and Public Health

Sincerely yours,

The authors

Reviewer 2 Report

This article seems to be a timely and informative report. However, it does not provide quantitative analysis to confirm or reject hypotheses.

Results shown here are rather descriptive and it may be helpful to add a table or tables to compactly summarize obtained information.

The interview has the demerit that the interviewer's opinion causes bias in the interviewee's response, and it is difficult for the interviewee to express a response against what society expects of him.

Although the computer-assisted analysis of the interview is an interesting method, it seems difficult to judge whether the derived conclusion, in particular the extraction of the three emerging themes, is sufficiently objective or depends on the software used for the analysis.

A disadvantage of this article is that no comparison has been made between physical therapists and other health professionals, or between COVID-19 and other epidemic crises. It is not entirely clear whether the authors believe their results are specific to physical therapists and different from other health care professionals, and whether the findings are specific to the management of COVID-19 patients or applicable to Ebola or to other patients threatened with death.

 As a whole, this article is well written and contains useful information on the mental response of physical therapists encountered with COVID-19, although the analysis does not appear to be fully standardized, and not quantitative.

As a quite minor point,

Line 51 COVID-190 should be COVID-19

Author Response

Response Letter manuscript IJERPH-1036271

Emotional Experience and Feelings during first COVID-19 Outbreak perceived by Physical Therapists: A Qualitative Study in Madrid, Spain

We thank the reviewers for their comments. We would like to make the Editor know that we addressed suggestions made by the reviewer one and answered those from reviewer two who it seems has not understood that this is qualitative, and not quantitate, study. We have included a new figure summarizing the themes found in the study which we believe will help reader. All changes are highlighted to facilitate the review process.

Reviewer 2

This article seems to be a timely and informative report.

Response: We would like to thank for this positive feedback.

However, it does not provide quantitative analysis to confirm or reject hypotheses.

Response: We apologize but we do not understand this comment since this is a qualitative exploratory study and not quantitative study as it is reflected in the title and throughout the text. In addition, the paper was submitted to a special issue of the journal focusing on qualitative research.

O'Brien et al (2014) reported that: “Reporting standards for titles, abstracts, and introductory material (problem formulation, research question) in qualitative research are very similar to those for quantitative research, except that the results reported in the abstract are narrative rather than numerical, and authors rarely present a specific hypothesis.” (p.1246)

As Creswell & Poth described (2018): “Qualitative research begins with assumptions and the use of interpretive/theoretical frameworks that inform the study of research problems addressing the meaning individuals or groups ascribe to a social or human problem. To study this problem, qualitative researchers use an emerging qualitative approach to inquiry, the collection of data in a natural setting sensitive to the people and places under study, and data analysis that is both inductive and deductive and establishes themes. The final written report (…) includes the voices of participants (…) a complex description of the problem…” (p.8)

In qualitative research there is research question but not research hypotheses. In qualitative research we do not seek to extrapolate the results to the general population (Creswell & Poth, 2018) but rather, the intention is to deepen our knowledge of people who suffer from illnesses or specific situations of health and illness (e.g. COVID-19).

 “The intent in qualitative research is not to generalize the information but to elucidate the particular, the specific.” (Creswell & Poth, 2018.p.158).

Carpenter and Suto (2008) regarding the research purpose and qualitative research question, described that: “The overall purpose of qualitative research can be broadly described as descriptive, exploratory, explanatory or emancipatory (…) Exploratory research refers to inquiry that breaks new ground by examining experiences, situations and meanings and, in doing so, identifies concepts previously ignored. Exploratory questions are typically broad, and researchers are motivated by a desire to find out more about what is happening, to seek new insights, to to asses phenomena from a new perspective.” (p.41)

Also, regarding qualitative research question and why should the research question be broad and open, Korstjens & Moser (2017) reported that: “To enable a thorough in-depth description, exploration or explanation of the phenomenon under study, in general, research questions need to be broad and open to unexpected findings (…) Where quantitative research asks: ‘how many, how much, and how often?’ qualitative research would ask: ‘what?’ and even more ‘how, and why?’ Depending on the research process, you might feel a need for fine-tuning or additional questions. This is common in qualitative research as it works with ‘emerging design,’ which means that it is not possible to plan the research in detail at the start, as the researchers have to be responsive to what they find as the research proceeds.” (p.275)

Reference:

  • Carpenter C, Suto M. Qualitative research for occupational and physical therapists: A practical guide. Oxford: Black-Well Publishing, 2008.
  • Creswell JW, Poth CN. Qualitative inquiry and research design. Choosing among five approaches. 4 ed. Thousand Oaks: SAGE, 2018.
  • Korstjens I, Moser A. Series: Practical guidance to qualitative research. Part 2: Context, research questions and designs. Eur J Gen Pract. 2017;23(1):274-279. doi:10.1080/13814788.2017.1375090
  • O'Brien BC, Harris IB, Beckman TJ, Reed DA, Cook DA. Standards for reporting qualitative research: a synthesis of recommendations. Acad Med. 2014;89(9):1245-51.

Results shown here are rather descriptive and it may be helpful to add a table or tables to compactly summarize obtained information.

Response: We believe that this reviewer has not understood that we have conducted a qualitative study, where results are descriptive as it has been presented. In fact, table 3 reflects a summary of the themes and categories (the results of the study). We have now included a new figure summarizing the themes found in the study which we believe will help readers to quickly visualize the results of the paper as suggested by the reviewer (lines 462-466).

Qualitative research does not mean that it is a type of research that uses qualitative variables that are quantifiable and analyzed (Creswell & Poth, 2018). Essentially, qualitative research mainly works with qualitative data such as transcribed narrative texts (obtained in the interviews), images (pictures, photographs), and written documents (letters, diaries).

“Qualitative data can be derived from interaction with participants, such as interviews or focus groups, or as a result of participant observation or analysis of documents or records. Qualitative data, whether in the form of transcripts or field notes, are generally presented in narrative form. When derived from interaction with participants the data are presented as verbatim quotations, to preserve and represent the voice of the participants.” (Carpenter & Suto, 2008, p.32)

Reference:

  • Carpenter C, Suto M. Qualitative research for occupational and physical therapists: A practical guide. Oxford: Black-Well Publishing, 2008.
  • Creswell JW, Poth CN. Qualitative inquiry and research design. Choosing among five approaches. 4 ed. Thousand Oaks: SAGE, 2018.

The interview has the demerit that the interviewer's opinion causes bias in the interviewee's response, and it is difficult for the interviewee to express a response against what society expects of him.

Response: We do not understand this statement. The realization of data collection in qualitative research requires interaction between researcher and participant. This does not mean that the researcher influences the participant to answer what they want to hear. That is why they are open questions, so that the participant can express himself without limitation.

The aim of qualitative interviews is to elicit the participant’s experiences, perceptions,

thoughts and feelings.

Moser & Korstjens (2018) described qualitative data tools as: “Interviews are another data collection method in which an interviewer asks the respondents questions, face-to-face, by telephone or online. The qualitative research interview seeks to describe the meanings of central themes in the life world of the participants. The main task in interviewing is to understand the meaning of what participants say (…) Data collection in qualitative research is unstructured and flexible. You often make decisions on data collection while engaging in fieldwork, the guiding questions being with whom, what, when, where and how.”(p.12)

Also, Moser & Korstjens (2018), regarding qualitative interviews, reported that: “Interviews involve interactions between the interviewer (s) and the respondent(s) based on interview questions. The interview questions are written down in an interview guide for individual interviews or a questioning route for focus group discussions, with questions focusing on the phenomenon under study. It should be a dialogue, not a strict question–answer interview. You are in control in the sense that you give direction to the interview, while the participants are in control of their answers. However, you need to be

open-minded to recognize that some relevant topics for participants may not have been covered in your interview guide or questioning route, and need to be added.” (p.13)

Also, Moser & Korstjens (2018) regarding face-to-face qualitative interviews reported that: “…face-to-face interview is an individual interview, that is, a conversation between participant and interviewer. Interviews can focus on past or present situations, and on personal issues. Most qualitative studies start with open interviews to get a broad ‘picture’ of what is going on. You should not provide a great deal of guidance and avoid influencing the answers to fit ‘your’ point of view, as you want to obtain the participant’s

own experiences, perceptions, thoughts, and feelings. You should encourage the participants to speak freely.” (p.13-14)

The Consolidated Criteria for Reporting Qualitative Research (COREQ) (Tong et al., 2007) reported that: “Data collection: The questions and prompts used in data collection should be provided to enhance the readers’ understanding of the researcher’s focus and to give readers the ability to assess whether participants were encouraged to openly convey their viewpoints.” (p.356).

References:

  • Moser A, Korstjens I. Series: Practical guidance to qualitative research. Part 3: Sampling, data collection and analysis. Eur J Gen Pract. 2018;24(1):9-18. doi:10.1080/13814788.2017.1375091
  • Tong A, Sainsbury P, Craig J. Consolidated criteria for reporting qualitative research (COREQ): a 32-item checklist for interviews and focus groups. Int J Qual Health Care. 2007;19(6):349–57.

Although the computer-assisted analysis of the interview is an interesting method, it seems difficult to judge whether the derived conclusion, in particular the extraction of the three emerging themes, is sufficiently objective or depends on the software used for the analysis.

Response: The present study is a qualitative study, where researchers want to know the personal and subjective experience of participants and not to get objective and quantitative data. In fact, as it is detailed in the text, we have followed the most restrictive procedures used in qualitative exploratory research.

This study was conducted according to Consolidated Criteria for Reporting Qualitative Research (COREQ) (Tong et al., 2007) and the Standards for Reporting Qualitative Research (SRQR) (O´Brien et al., 2014). The credibility of the findings can be assessed if the process of coding (selecting significant sections from participant statements), and the derivation and identification of themes. Researchers sometimes use software packages to assist with storage, searching and coding of qualitative data. But is an option (Tong et al., 2007). In the present study, we reported that no qualitative software was used to analyze the data.

Moser & Korstjens (2018) showed that: “An inductive analysis involves breaking down the data into smaller units, coding and naming the units according to the content they present, and grouping the coded material based on shared concepts.” (p.17)

Also, Moser & Korstjens (2018) regarding analysis reported that: “In general, qualitative analysis begins with organizing data. Large amounts of data need to be stored in smaller and manageable units, which can be retrieved and reviewed easily. To obtain a sense of the whole, analysis starts with reading and rereading the data, looking at themes, emotions and the unexpected, taking into account the overall picture. You immerse yourself in the data. The most widely used procedure is to develop an inductive coding scheme based on actual data (…) Based on this close examination of what emerges from the data you make as many labels as needed. Then, you make a coding sheet, in which you collect the labels and, based on your interpretation, cluster them in preliminary categories. The next step is to order similar or dissimilar categories into broader higher order categories. Each category is named using content-characteristic words. Then, you use abstraction by formulating a general description of the phenomenon under study: subcategories with similar events and information are grouped together as categories and categories are grouped as main categories. During the analysis process, you identify ‘missing analytical information’ and you continue data collection. You reread, recode, reanalyse and re-collect data until your findings provide breadth and depth.”(p.16)

References:

  • Moser A, Korstjens I. Series: Practical guidance to qualitative research. Part 3: Sampling, data collection and analysis. Eur J Gen Pract. 2018;24(1):9-18. doi:10.1080/13814788.2017.1375091
  • Tong A, Sainsbury P, Craig J. Consolidated criteria for reporting qualitative research (COREQ): a 32-item checklist for interviews and focus groups. Int J Qual Health Care. 2007;19(6):349–57.
  • O'Brien BC, Harris IB, Beckman TJ, Reed DA, Cook DA. Standards for reporting qualitative research: a synthesis of recommendations. Acad Med. 2014;89(9):1245-1251.

A disadvantage of this article is that no comparison has been made between physical therapists and other health professionals, or between COVID-19 and other epidemic crises.

Response: Qualitative research has different viewpoint of the experience than quantitative research; therefore, comparison between this unprecedent crisis caused by the COVID-19 and the lack of previous qualitative research on physical therapists make not possible to compare these results with previous epidemic crisis.

Nevertheless, we do not completely understand this comment from the reviewer since the discussion section summarizes the experiences reported by other health care professionals who had worked with COVID-19 patients on first line. We have included some sentences comparing the results on physical therapists with other healthcare professionals (always considering that qualitative research should be prudently compared) in the discussion (lines 478, 482, 508-510, 516-522, 530-532, 546, 552-553, 556, 558, 587-589, 598, 605-609).

It is not entirely clear whether the authors believe their results are specific to physical therapists and different from other health care professionals, and whether the findings are specific to the management of COVID-19 patients or applicable to Ebola or to other patients threatened with death.

Response: The lack of previous qualitative studies assessing the experience of physical therapists in other situations such as Ebola makes extremely difficult, if not impossible, to extrapolate current findings to other situations. In fact, qualitative research reflects the most personal experience of people, therefore, extrapolation to different situations is not generally recommended. Nevertheless, we have included some sentences comparing the results on physical therapists with other healthcare professionals (always considering that qualitative research should be prudently compared) in the discussion (lines 478, 482, 508-510, 516-522, 530-532, 546, 552-553, 556, 558, 587-589, 598, 605-609).

 As a whole, this article is well written and contains useful information on the mental response of physical therapists encountered with COVID-19, although the analysis does not appear to be fully standardized, and not quantitative.

Response: Again, we would like to thank for this positive feedback from the reviewer and we hope that explanations that we conducted a qualitative, and not quantitative, study and also that the paper was submitted to a special issue of the journal focusing on qualitative research are helpful.

Furthermore, the authors have included specific information on the qualitative analysis process in previous questions.

As a quite minor point,

Line 51 COVID-190 should be COVID-19

Response: Thanks. This typo has been edited

We hope that the current version of the manuscript can get a positive review and can be accepted for publication in International Journal of Environmental Research and Public Health

Sincerely yours,

The authors
